# Evaluating the Effects of Some Selected Medicinal Plant Extracts on Feed Degradability, Microbial Protein Yield, and Total Gas Production In Vitro

**DOI:** 10.3390/ani13040702

**Published:** 2023-02-17

**Authors:** Aderonke N. Abd’quadri-Abojukoro, Ignatius V. Nsahlai

**Affiliations:** Discipline of Animal and Poultry Science, University of KwaZulu-Natal, Private Bag X01, Scottsville 3201, South Africa

**Keywords:** plant extracts, incubation period, degradation, microbial yield, gas production

## Abstract

**Simple Summary:**

Ruminant animals are major producers of animal protein, and their uniqueness in utilizing unconventional feed resources cannot be overemphasized. In the tropical environment, most of the unconventional feeds available for this class of animals are of low quality and are characterized by high structural carbohydrates which lack adequate fermentable carbohydrates and nitrogen composition, a factor of crude protein. Feeding poor-quality forage has been linked to higher methane production from ruminants, and improving the rumen fermentation of these poor forages can improve animal performance and reduce methane emission. Many strategies have been employed in recent times, and the most successful is the use of antibiotics, but there is a campaign against the use of antibiotics. Other strategies such as the use of natural products such as plant extracts rich in antibiotics and phytochemicals have been exploited. This study evaluates the effect of phytochemical-rich plant extracts on fermentation parameters in vitro, and the results were promising.

**Abstract:**

This study evaluates the effect of 22 crude ethanolic plant extracts on in vitro rumen fermentation of *Themeda triandra* hay using monensin sodium as a positive control. The experiment was run independently three times at 16 and 48 h of incubation periods using the in vitro gas production techniques. Fermentation parameters were determined at both hours of incubation. Plant extracts influenced gas production (GP) in a varied way relative to control at both hours of incubation, and GP is consistently highly significant (*p* < 0.0001) at 16 and 48 h. Microbial protein yield (MY) was not significantly affected at 16 h (*p* > 0.05), but it was at 48 h (*p* < 0.01). Higher MY was recorded for all treatments except for *A. sativum* and *C. intybus* at the early incubation stage (16 h) relative to 48 h of incubation. Compared to the control group at 48 h, all plant extracts have higher MY. After 48 h of incubation, the result shows that plant extracts have an effect on fermentation parameters determined; ruminal feed degradation, gas production, microbial protein yield, and partitioning factor in varied manners. All the plant extracts improve the MY which is the major source of amino acids to ruminants and has significant importance to animal performance. *C. illinoinensis*, *C. japonica, M. nigra, P. americana, C. papaya*, and *A. nilotica* (pods) were the most promising plant extracts, but further study is recommended to validate the in vitro observation in vivo.

## 1. Introduction

Ruminant animals are major producers of animal protein, and their uniqueness in utilizing feed resources that cannot be used by other class of animals and humans affords them an advantage that can hardly be overestimated. The ever-increasing human population over the years and the need to meet their animal protein demand have led to increased ruminant farming across the globe [1]. These animals are essential not only for their excellent quality protein but also as draft animals as well as for providing raw material for industries such as wool, leather, tallow, and bone, among others [1].

The unavailability of quality feed is hampering livestock productivity in most developing countries. Ruminant production in the tropics and developing countries depend solely on natural/native pastures, in this study Themeda trianda will be used as a case study, and it is an example of natural pasture of low quality which is readily available for ruminants in this part of the world. They are often only supplemented with agro-industrial roughages during the dry season when forages are limited. These are characterized by high structural carbohydrates (ligno-celluloses) that lack adequate fermentable carbohydrates and nitrogen composition, a factor of crude protein [2]. A kind of feeding system that has been associated with low digestibility and voluntary intake, which largely limits animal growth and productivity [3]. Moreover, feeding poor quality forage has been linked to higher methane production from ruminants [4], which represent a considerable loss of gross energy intake and digestible nutrient by animals [5,6]. Methane emission is now of global interest because of its effect on climate change with its higher global warming potential relative to carbon dioxide [7].

It is crucial to emphasize the importance of rumen fermentation on ruminants and their health because it may invariably have a potential influence on humans. This influence may come through the nutritional quality of their products and the environment; through the impact of the emission of greenhouse gases, and excessive excretion of nitrogen in their feces and urine [8]. The end products of rumen fermentation are characterized by the production of VFA and ammonia for microbial protein synthesis, which are the ruminant animal’s primary source of energy and amino acids, respectively [9]. Of importance is the production and emission of gases, which are nutritionally wasteful and environmental pollutants, that are of great concern [3,9].

Improving rumen fermentation is one of the goals in ruminant production because efficient rumen fermentation will lead to increase productivity, which most times bring about a decrease in methane (a potent greenhouse gas). Various approaches to improving the productivity of these classes of animals have been established; from feeding strategy to breed selection, and microbial ecology selection (such as defaunation), to the use of antibiotic ionophores, synthetic/organic compounds, other chemical additives, and natural compounds. The general aim of all these strategies is to manipulate the rumen microbial ecosystem towards enhanced fermentation, and increased fiber digestibility and microbial protein synthesis while mitigating methane production and emission from ruminants [10]. In their editorial, Ungerfeld and Newbold [8] reiterate that for a more productive and sustainable ruminant production, manipulating rumen microbial activity is ineludible.

The campaign against the use of antibiotics in livestock production has gained momentum, especially after the ban of antibiotics used in animal production in the EU, and this has led to the search for a natural alternative to antibiotics for animal husbandry. One of the alternatives to antibiotics suggested by Seal et al. [11] at a symposium is the use of phytonutrients (plant and plant extracts). Earlier, Greathead [12] emphasized the use of plants and plant extracts and give several possible ways of exploiting them for improving animal production, particularly in ruminants as regards rumen fermentation and metabolism.

Several researchers have reported [13,14,15,16,17,18,19] that plants rich in phytochemicals and their extracts are promising alternatives to antibiotics and chemical additives as rumen modifiers. These plant products could bring about improved fermentation efficiency, mitigate methane emissions, and may probably improve animal productivity. Unfortunately, most of this research does not pay much attention to the antibacterial activity and cytotoxicity aspect of the plant extracts used in their study. This study is part of a project, and the antibacterial and cytotoxicity report of all plant extracts used in this study has been reported [20,21].

The aim of this study was to evaluate the effect of some selected medicinal plant extracts with known bioactivity as a source of rumen microbial ecosystem modifiers, for their influence on dry matter and fiber degradation, microbial protein synthesis, and total gas production in vitro. While the hypothesis of the study is that medicinal plant extracts have an effect on rumen fermentation end products in vitro.

## 2. Materials and Methods

### 2.1. Collection of Plant Materials

All the plant materials used in this study (Table 1) were collected from the University of KwaZulu-Natal botanical garden, Pietermaritzburg with geographical coordinates 29°37′ S and 30°24′ E at an altitude of 659 m and mean annual rainfall of 695 mm, and at the Ukulinga research farm of University of KwaZulu-Natal (UKZN), Pietermaritzburg, with 29°39′ S and 30°24′ E at an altitude of 700 m and mean annual rainfall of 735 mm. Garlic, ginger, and onion samples were purchased from a commercial supermarket. While *Persia americana*, *Vernonia amygdalina*, and *Psidium guajuava* leaves were collected from a private residence around UKZN, Pietermaritzburg campus. All plant parts used were properly identified and confirmed with the help of a botanist from the Department of Botany, University of KwaZulu-Natal, Pietermaritzburg.

### 2.2. Preparation of Plant Extracts

All the plant materials (leaves) were washed with tap water immediately after collection and air-dried, while the nutshell and pods were sorted. All plant extracts were prepared as described by Abd’quadri-Abojukoro et al. [20,21]. Briefly, all plant materials were chopped into smaller pieces, and they were all oven-dried at 40 °C until they were completely driy. The dried plant materials were milled to pass through a 1 mm sieve, and they were extracted in a Soxhlet apparatus with 80% ethanol as the solvent (in a 1:10 of milled plant material to the solvent) until a clear solvent was seen around the thimble in the extraction chamber. The extracts were concentrated in the water bath at 60 °C to determine the extract yield and then reconstituted to a concentration of 50 mg/mL.

### 2.3. Chemical Analysis of Substrate

A sample of the substrate, *Themeda triandra* hay, was collected from the livestock section of the Ukulinga research farm. The feed sample was analyzed for its dry matter (DM) and total ash according to the Association of Official Analytical Chemists (AOAC). Crude protein (CP) was determined by analyzing the nitrogen concentration of the feed with LECOTruMac CNS analyser (LECO TruMac Series Macro Determinator, LECO Corporation, Saint Joseph MI, USA). The nitrogen concentration was multiplied by 6.25. Neutral detergent fiber (NDF) and acid detergent fiber (ADF) were determined according to Van Soest et al. [22] using ANKOM 200/220 fiber analyzer (ANKOM Technology, Fairport, NY, USA). The subtraction of ADF from NDF was used to determine hemicellulose.

### 2.4. Preparation of Incubation Medium

McDougall’s salivary buffer solution was prepared as described by Basha et al. [23]; the buffer solution was prepared from two different solutions (solutions A and B). Solution A was prepared by dissolving 19.6 g of NaHCO_3_, 7.4 g of NaHPO_4_, 1.14 g of KCl, 0.94 g of NaCl, and 0.26 g of MgCl.6H_2_O into 2 L of distilled water, whereas for Solution B, 2.65 g of CaCl_2_.2H_2_O was dissolved in 50 mL of distilled water. The complete salivary buffer was prepared by adding 2 mL of solution B into 2 L of solution A drop-wise while stirring, and 5.6 g of (NH_4_)_2_SO_4_ was added to the mixture. The solution was then warmed to 39 °C in a water bath with continuous stirring and saturated with CO_2_.

Rumen liquor was collected before morning feeding from three ruminally fistulated jersey cattle which had free access (grazing) to Kikuyu pasture (*Pennisetum clandestinum*). The grazing was supplemented with urea-treated *Themeda triandra* hay, and the cattle were offered freshwater and salt lick ad libitum. Rumen liquor was filtered through four layers of cheesecloth into a pre-warmed flask (39 °C) that has was flushed with CO_2_ and was transported to the laboratory immediately where continuous CO_2_ flushing continued until the incubation began. The final buffer and rumen liquor mixture were in a 2:1 ratio.

### 2.5. In Vitro Rumen Fermentation of Themeda triandra Hay

The substrate used was *Themeda triandra* hay (veld hay), and it was dried at 60 °C for 72 h and milled to pass through a 1 mm screen. A 250 mL Duran bottle was used for the fermentation of 1000 mg of the weighed substrate. One milliliter of the 22 crude plant extracts (5% *w*/*w* of substrate mass) was pipetted into each bottle separately. Monensin was added at the rate of 100 ppm per bottle which serves as the positive control, while 1 mL of distilled water was pipetted into the negative control bottle, and three blanks (only buffer and rumen fluid) were included per run. Sixty-seven milliliters of the prepared McDougall’s buffer solution was added to each bottle (22 plant extracts, positive and negative controls, and three blanks). All bottles were kept in a water bath at 39 °C for one hour to allow soaking of the substrate and plant extract where applicable before inoculation (addition of rumen fluid). Later, when rumen fluid was brought, 33 mL of the rumen fluid was added to each bottle to complete the inoculation under the CO_2_ stream. The bottles were then sealed immediately with their lids and tightened to avoid gas escape during incubation.

The sealed Duran bottles were then incubated in a previously conditioned incubator (39 °C) with an oscillatory motion stirrer at 120 rpm set at 2 min intervals and channels of pressure sensors were fitted. This entire process was repeated three times of independent run for incubation periods of 16 h and 48 h.

Total gas production was determined after each incubation period; the pressure transducer that has been connected and programmed on the computer with a digital data logger was connected to the pressure channels on each bottle. The pressure logged at the end of the 16 h and 48 h incubation period was used to determine the total gas production by converting the gas pressure to volume. *Gas* pressure readings were displayed on the computer in millibars and pascal. The gas pressure was converted to gas *Volume* (mL) using Boyle’s gas law relationship, as reported by Mauricio et al. [24].
(1)$$Gas\;Volume=\frac{Vh}{Atm}\times Actual\;pressure$$
where *Vh* is the headspace volume of the Duran bottles (mL), *Atm* is the atmospheric pressure (millibars), *Actual pressure* is the pressure read by the transducer as displayed on the computer in millibars, and the gas volume was corrected for blanks.

At the end of each incubation period (16 and 48 h), the pH was measured using a CRISON Micro pH 2000 (CRISON Instruments, Barcelona, Spain). Fermentation bottles were then placed on ice to stop further fermentation. The cultured solution from each Duran bottle was transferred into centrifuge tubes and centrifuged at 10,000 rpm for 20 min at 4 °C, and the supernatant was discarded. The residue was transferred into an aluminum foil container with known weight and placed in the oven to dry at 80 °C for 2–3 days or until a constant weight was achieved (completely dried). The dried residues were corrected for blank incubation, that is, incubation bottles that contained only buffer and rumen fluid. The difference in mass incubated and mass of dry residue gives apparent degradability (*Apdeg*). The residue was then refluxed with the neutral detergent solution using ANKOM 200/220 fiber analyzer. The resultant *NDF* mass after correction for blanks was used to determine true degradability (*Trdeg*) according to Van Soest et al. [22], and microbial yield was determined according to Van Soest [25] and Blummel et al. [26] as follows:(2)Trdeg=mass of substrate incubated−mass of residue after NDF treatment
(3)MY=Trdeg − Apdeg

*NDF* degradability (*NDFD*) was calculated using the following formula, according to Goering and Van Soest [27]:(4)$$NDFD\left(g/\mathrm{KgDM}\right)=\frac{1000\times \left(NDFsubstrate-NDFresidue\right)}{NDFsubstrate}$$

While the partitioning factor (PF) was calculated by dividing the Trdeg by volume of gas produced at the end of each incubation period [26].

### 2.6. Statistical Analysis

All data generated from the experiment for the two incubation periods, 16 h and 48 h, were fitted into the analysis of variance using the general linear model (GLM) procedure of SAS, and the difference between means was tested using the Tukey test [28].
Statistical model: Y_ij_ = µ + T_i_ + ꞵ_j_ + Ɛ_ij_
where, Y_ij_ = observation k in treatment i, µ = overall mean, T_i_ = treatments effect of ith (plant extracts and monensin), ꞵ_j_ = effect of run jth (block), and Ɛ_ij_ = residual error with mean 0 & variance σ^2^.

## 3. Results

### 3.1. Chemical Composition of the Substrate (Themeda triandra Hay)

The chemical composition of the substrate used for evaluating the effect of medicinal plant extracts on in vitro fermentation shows that the hay used as the substrate is of low quality even after it has been improved with urea. *Themeda triandra* hay as substrate contained (g/Kg dry matter basis) 56 CP, 739 NDF, 556 ADF, 183 HEM, 904 OM, and 96 Ash.

### 3.2. Dry Matter Degradability, Microbial Protein Yield and Gas Production of Themeda triandra Hay

In vitro degradability parameters (Apdeg, Trdeg, NDFD) and gas production after 16 h of incubation were all influenced by the inclusion of all the plant extracts and monensin (Table 2). These additives affected Apdeg (*p* < 0.01), Trdeg and NDFD (*p* < 0.05), and gas production (*p* < 0.0001), while microbial yield was not affected (*p* > 0.05) by the additives at 16 h of fermentation. Inclusion of all plant extracts and monensin causes a decrease in Apdeg relative to control. However, this reduction was highly significant (*p* < 0.01) with *C. illinoinensis* having a negative value (−16 g Kg^−1^ DM). In comparison to other treatments with varying Apdeg, the control treatment (255 g Kg^−1^ DM) had the highest Apdeg. However, samples containing *T. violacea* had the highest Trdeg and NDFD (511 g Kg^−1^ DM and 250 g Kg^−1^ DM, respectively), and the lowest Trdeg and NDFD were observed with *C. illinoinensis* (407 g Kg^−1^ DM and 146 g Kg^−1^ DM, respectively) plant extract.

Gas production by all the additives after 16 h of fermentation varied widely. However, it was not statistically different in comparison to means of control (42 mL g^−1^) except for samples containing *C. illinoinensis* (6 mL g^−1^) plant extract with the least gas production. Plant extract of *C. intybus* (62 mL g^−1^) had the highest gas production. The inclusion of monensin sodium (13 mL g^−1^) significantly (*p* < 0.0001) reduced gas production relative to the plant extracts of *C. intybus*, *C. limon*, *V. amygdalina*, *A. sieberiana*, and *Z. officinale*.

After 48 h of incubation, all the observed parameters (Table 3) were significantly influenced by the treatments; Apdeg (*p* < 0.0001), Trdeg and NDFD (*p* ≤ 0.05), MY (*p* < 0.01), and gas production (*p* < 0.0001). Apdeg was highest in samples containing *C. limon* (385 g Kg^−1^ DM) plant extracts and lowest in *C. illinoinensis* samples (46 g Kg^−1^ DM), which were significantly (*p* < 0.0001) different from all other treatments except *P. guajuava* and *V. amygdalina*.

The addition of crude extracts of *C. papaya* leaves had the highest Trdeg and NDFD (587 g Kg^−1^ DM and 326 g Kg^−1^ DM, respectively), although it did not differ significantly from all other treatments except *C. illinoinensis* with the least Trdeg and NDFD (484 g Kg^−1^ DM and 223 g Kg^−1^ DM, respectively). Unlike after 16 h of fermentation, the means of microbial yield after 48 h of fermentation differed significantly (*p* < 0.01) among extracts; all samples with plant extracts and monensin had higher microbial yield than the control which ranges from 438 g Kg^−1^ DM—151 g Kg^−1^ DM. *C. illinoinensis* plant extract had the highest microbial yield which differed significantly (*p* < 0.01) from samples containing *C. arabica*, *A. comosus*, *A. ferox*, *M. nigra*, *A. cepa*, *A. sieberiana*, *C. limon*, and control.

The addition of plant extracts and monensin sodium to the hay influences the total gas production after 48 h of fermentation significantly (*p* < 0.0001). Plant extracts of *C. limon* (140 mL g^−1^ DM incubated) had the highest gas production. In contrast, treatments containing plant extracts of *P. americana*, *M. nigra*, *A. nilotica* (pod), *C. illinoinensis*, *C. japonica*, and monensin sodium (40, 44, 53, 62, 64, and 66 mL g^−1^ DM incubated) reduced gas production significantly (*p* < 0.0001) relative to *C. limon* extract but not with the control (104 mL g^−1^ DM incubated).

The pH values observed after 16 h of incubation (Table 4) were significantly (*p* < 0.01) affected by the inclusion of plant extracts and monensin sodium. The effects of the addition of kernel shell extract of *C. illinoinensis* and monensin sodium were different from *F. natalensis* and *T. violacea* plant extracts for their pH after 16 h of incubation. Means of all the treatments were like the control for their pH value statistically, and the pH ranges from 6.96 (*C. illinoinensis)* to 6.80 (*T. violacea*). Similarly, the pH values of all the treatments were not different relative to the control after 48 h of incubation; the pH ranging from 6.91 (*C. illinoinensis*) to 6.60 (*M. oleifera*). Overall, *C. illinoinensis* plant extract still maintained the highest pH, which differs significantly (*p* < 0.05) from the means of *T. violacea* and *M. oleifera* plant extracts.

The inclusion of *C. intybus* plant extract brought about the least PF (7.8 mg mL^−1^) which is not statistically different from the control (11.1 mg mL^−1^). Meanwhile, the influence of *C. illinoinensis* and monensin sodium on the PF differed significantly (*p* < 0.0001) from each other and all other plant extracts including the control after 16 h of fermentation. The partitioning factor also differed (*p* < 0.0001) amongst extracts at 48 h of incubation. *P. americana* (14.32 mg mL^−1^) had the highest partitioning factor, which differed significantly from all other treatments and control except *M. nigra*, *C. japonica*, and *A. nilotica* (pods) (12.56, 9.79, and 10.56 mg mL^−1^, respectively). While the least PF was obtained from the plant extract of *C. limon* (4.04 mg mL^−1^) which did not differ statistically from the control. Figure 1 and Figure 2 represent the relative effect of the treatment (plant extracts) to control on Trdeg, MY and GP.

## 4. Discussion

The use of in vitro gas production techniques for the evaluation of the effect of plant and plant extracts containing phytochemicals on rumen fermentation has been considered acceptable [29]. It is important to indicate that all the plant extracts used in this study have been previously evaluated for their phytochemical composition quantitatively, cytotoxicity, and antibacterial activity by these authors [20,21]. In the present study, we endeavor to determine the effects of selected medicinal plant extracts on rumen fermentation parameters at 16 h and 48 h of incubation. It has been suggested that it is crucial to identify the incubation time in the in vitro gas technique at which the partitioning factor (a measure of the fermented substrate that leads to microbial mass synthesis) is maximum [3] to determine when it should be measured for accurate determination in an in vitro system [30]. This was in resonance with the findings of Ouda and Nsahlai [31] who stated that when using the in vitro gas technique to estimate the dynamics of feed degradation and protein metabolism, the correct timing of incubation should be considered. It was observed in this study that the microbial protein yield for all treatments was higher at 16 h of incubation when compared to yields at 48 h. This agreed with the observation of Getachew et al. [32], who reported that microbial growth efficiency was higher for 16 h of incubation than 24 h in their study on tannin-rich feeds. Among the essential end product of rumen fermentation is the synthesis of microbial protein, which is the primary source of protein supply post-ruminally, especially in animals fed high-fibrous feed.

However, reduced microbial yield observed in this study after 48 h of fermentation as compared to 16 h of fermentation can probably be a result of incomplete fermentation of the neutral detergent solubles in the substrate at an early stage of incubation, energy spilling, or microbial lysis at the later stage of incubation. It is important to note that plant extracts with high efficiency of microbial protein yield will lead to efficient utilization of feed nitrogen and carbon [33]. In this study, we observed that the addition of all plant extracts at 48 h of incubation improves the microbial yield relative to the control. Moreover, improved microbial biomass yield observed in the current study might also be associated with decreased bacteria proteolysis that usually occurs in the presence of protozoans, since most phytochemicals present in these plant extracts are capable of defaunation [13]. Newbold et al. [34] in their meta-analysis confirmed the theoretical expectation that defaunation may lead to higher microbial biomass yield.

Post ruminal flow of microbial protein may result in the availability of protein for absorption in the small intestine [35]. The selection of plant extracts that improve microbial protein yield along with higher dry matter and fiber degradability will lead to a higher supply of protein post-ruminally. Higher microbial protein yield may decrease the flow of carbon in feed to fermentative CO_2_ and CH_4_. Due to microbial protein’s high biological value (high-quality protein), they serve as a good source of metabolizable protein for ruminants [36], which may reduce the need for feed supplementation of rumen undegradable protein and reduce the cost of feeding [33].

*C. illinoinensis* nutshell extract had a negative Apdeg at 16 h of fermentation; this can be explained from its phytochemical composition point of view [21]. A higher concentration of condensed tannins in feed samples is known for its ability to cause nutrient precipitation and suppress digestibility [37]. Tannins might have precipitated the nutrient in the substrate fermented in the *C. illinoinensis* plant extract, making the nutrient unavailable for degradation, slowing down degradation, and thus leading to a negative balance of apparent degradation at 16 h of incubation.

In vitro NDF degradability has been indicated as an essential measure to accurately predict feed intake, total digestible nutrients, and net energy of forages [38]. An increased in vitro NDF degradability by most plant extracts is an indication of their potential to cause an increased forage intake and digestible energy when fed as an additive to ruminants. In the current study, the ethanolic leaf extract of *C. papaya* was able to improve in vitro NDF degradability by up to 20% relative to control, while monensin improves it by 11% at 48 h of incubation. It has been established that a one-unit increase in in vitro NDF digestibility is associated with up to a 17% increase in dry matter intake and over 24% increase in fat-corrected milk yield [38].

True degradability of feed or substrate in vitro can only lead to either gas production or fixed into microbial cells for microbial biomass synthesis [39]. The Trdeg, TGP, and microbial biomass yield observed as a result of the addition of *C. illinoinensis* plant extract to the *Themeda triandra* hay fermentation showed that the Trdeg is channeled more towards microbial protein synthesis, which was reflected in gas production, especially at 16 h of incubation. The reduced fiber degradation by the addition of *C. illinoinensis* and some other plant extracts may be associated with the effect of the plant extracts on protozoans (defaunation). Because it is well documented that defaunation reduces fiber digestion [40], and although some plant extracts rich in phytochemicals, especially saponins are capable of defaunation, their activity varies with type [41]. Also, it has been reported that saponin encourages cell lyses, which occur at saponin-cholesterol-rich domains, by penetrating into the lipid bilayer of the protozoan membrane and binds with cholesterol forming a micelle-like aggregation that leads to increased cell permeability [42].

Suppression of in vitro degradability of feed by the addition of plant extracts has been reported by Patra et al. [43], where they show that all plant extracts used in their study reduce dry matter degradation relative to control. This was not true with our observation with Trdeg, where 18 plant extracts were able to improve Trdeg by at least 1% relative to the control, which is an indication that most of the plant extracts were not detrimental to the rumen microbes responsible for digestion. In this context, our result resonates well with that of Akanmu and Hassen [17] where all plant extracts used in their study improved in vitro organic matter degradability. *C. papaya* leaf extracts, with the highest Trdeg, cause an improvement that is above 10% relative to control and have been reported to have some compounds (such as papain, chymopapain, and proteolytic enzymes) that aid digestion [44].

Plant extracts that are rich in tannins have been reported to reduce gas production [32] which was observed with some plant extracts in this study; for instance, *C. illinoinensis* and *C. japonica* with higher tannins were able to reduce gas production. Plant extracts that are lower in tannins such as *C. limon*, *Z. officinale*, and *A. sativum*, have higher gas production. Noteworthy is the effect of alkaloids on rumen fermentation in vitro. Makkar [29] stated that the inclusion of different alkaloids would cause a substantial decrease in Trdeg and gas production but a relatively lower decrease in microbial biomass synthesis. This agreed with the findings of this study where *P. guajava* and *C. illinoinensis* had the highest concentration of alkaloids as reported by Abd’quadri-Abojukoro et al. [21] and decreased gas production and Trdeg relative to control, although unlike in their report, *P. guajava* did not decrease microbial biomass yield. This finding is not consistent across all plant extracts and can be attributed to the interplay among several phytochemicals. Oskoueian et al. [45] highlighted that the presence of many compounds or metabolite in crude plant extract as used in this study may influence the result and makes it difficult to correlate the response to a particular phytochemical.

Rumen pH is an essential aspect of ruminant life and health because an extreme on the upper or lower limit (above 7.0 and below 5.5, respectively) can be detrimental to rumen microbes and rumen health. In the present study, the pH value recorded was within the normal range (6.0–7.0) for fiber and protein degradation [46]. This suggests that even with the addition of these plant extracts, cellulolysis and cellulolytic bacteria growth was not affected. It is noteworthy, to indicate that the higher pH observed at the earlier stage of fermentation (16 h) as compared to at 48 h can be explained from the standpoint of their antibacterial activity that has been established in the previous study on this same set of plant extracts [20]. All the plant extracts used in this study have antibacterial activity, and this might have affected the microbial activity at the earlier stage of fermentation which causes the higher pH observed.

Partitioning factor, which is the ratio of a substrate that was truly degraded to the gas produced in in vitro rumen fermentation and a measure of the efficiency of microbial protein yield. According to Blummel et al. [26], the theoretical range of PF is 2.75–4.41 for most feed, whereas the PF observed in this study ranged from 4.04 to 14.32 at 48 h of incubation. Our observation agrees with that of Getachew et al. [32] who established that PF for phytochemical (tannins) rich feeds ranged from 3.1 to 16.1. Furthermore, this can be due to the inhibitory effect of plant extracts and their phytochemicals on gas production, which caused a lower gas production per unit of a truly degraded substrate. The decrease in gas production observed in the current study by the addition of *P. americana* after 48 h of fermentation may be associated with its carminative properties [44], which prevent the formation of gas or gas buildup in the gastrointestinal tract and is responsible for the higher PF recorded. Despite the lower gas produced by *P. americana*, feed degradation was not affected negatively, when compared to the control. Higher PF implies that more degraded dry matter was incorporated into microbial biomass synthesis, which is an indication of improved fermentation efficiency and can lead to higher dry matter intake in vivo.

Generally, the variations observed in the effect of the different plant extracts can be due to differences in their phytochemical concentration and the chemical characteristic of the individual phytochemicals. Monensin sodium has been reported to have the ability to improve rumen fermentation and feed efficiency [47]. In our study, we observed that the addition of monensin improves dry matter Trdeg, NDF degradability, and microbial biomass yield, and it causes a reduction in gas production relative to control. Most plant extracts in this study were able to demonstrate similar to even better effects on those parameters. Akanmu et al. [19] reported that all plant extracts used in their study performed better than monensin.

## 5. Conclusions

All the plant extracts investigated improve microbial protein yield which is the major source of amino acids to ruminants and has significant importance to animal performance. Some of the plant extracts studied are promising and could improve the digestibility and utilization of poor forages which may eventually mitigate methane production from ruminants fed poor forages. However, promising plant extracts, such as *C. illinoinensis*, *C. japonica*, *M. nigra*, *P. americana*, *C. papaya*, and *A. nilotica* (pods) needed to be logically selected for further study in vivo to verify in vitro observation.

## Figures and Tables

**Figure 1 animals-13-00702-f001:**
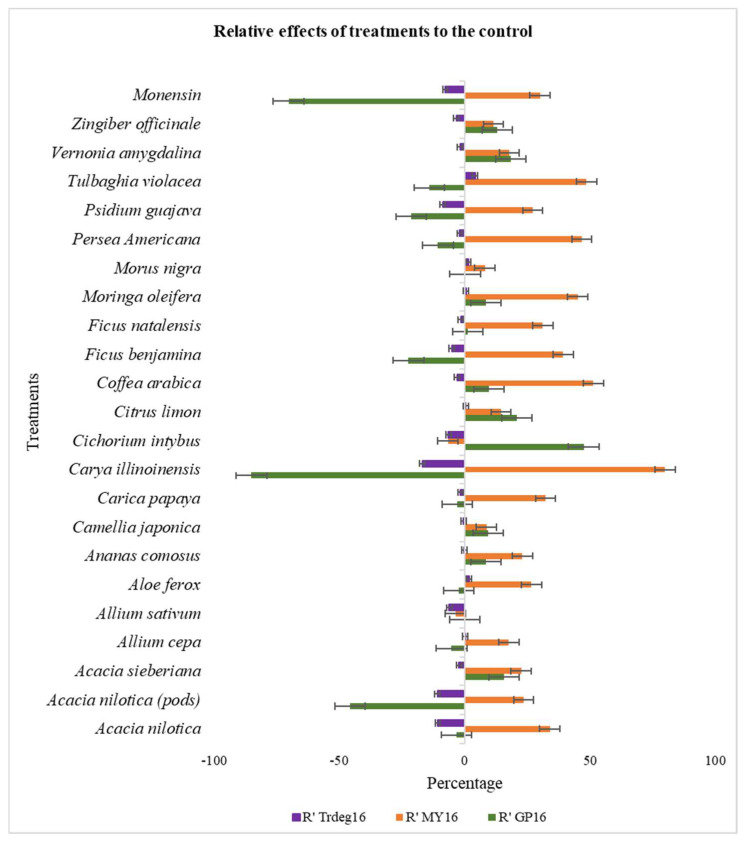
Chart presenting the effects of the treatments (plant extracts and monensin sodium) relative to the control on true degradability, microbial yield, and gas production at 16 h of incubation of *Themeda triandra* hay. R’ Trdeg—effect on true degradability, R’MY—effect on microbial yield, R’ GP - effect on gas production.

**Figure 2 animals-13-00702-f002:**
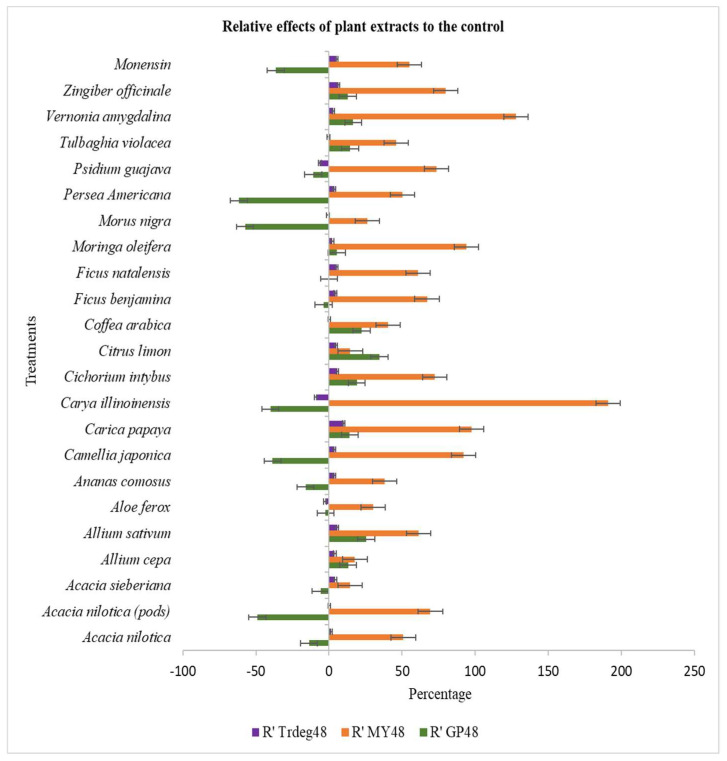
Chart presenting the effects of the treatments (plant extracts and monensin sodium) relative to the control on true degradability, microbial yield, and gas production at 48 h of incubation of *Themeda triandra* hay. R’ Trdeg—effect on true degradability, R’MY—effect on microbial yield, R’ GP- effect on gas production.

**Table 1 animals-13-00702-t001:** List of plant species evaluated for their effect on rumen fermentation in vitro.

Scientific Name	Common Name	Family Name	Part Used
*Acacia nilotica* L.	Gum Arabic	Fabaceae	Leaves
*Acacia nilotica* L.	Gum Arabic (pod)	Fabaceae	Pods with seeds
*Acacia sieberiana* DC.	Paperbark	Fabaceae	Leaves
*Allium cepa* L.	Onions	Liliaceae	Bulbs
*Allium sativum* L.	Garlic	Liliaceae	Bulbs
*Aloe ferox* Mill.	Aloe	Asphodelaceae	Leaves
*Ananas comosus* (L) Merr.	Pineapple	Bromeliaceae	Leaves
*Camellia japonica* L.	Tea	Theaceae	Leaves
*Carica papaya* L.	Pawpaw	Caricaceae	Leaves
*Carya illinoinensis* K.Koch	Pecan	Juglandaceae	Kernel shell
*Cichorium intybus* L.	Chicory	Asteraceae	Leaves
*Citrus limon* (L.) Osbeck	Lemon	Rutaceae	Leaves
*Coffea arabica* L.	coffee	Rubiaceae	Leaves
*Ficus benjamina* L.	Weeping fig	Moraceae	Leaves
*Ficus natalensis* Hochst.	Natal fig	Moraceae	Leaves
*Moringa oleifera* Lam.	Drumstick	Moringaceae	Leaves
*Morus nigra* L.	Mulberry	Moraceae	Leaves
*Persea Americana* Mill.	Avocado	Lauraceae	Leaves
*Psidium guajava* L.	Guava	Myrtaceae	Leaves
*Tulbaghia violacea* Harv.	Wild garlic	Alliaceae	Whole plant
*Vernonia amygdalina* Delile	Bitter leaf	Asteraceae	Leaves
*Zingiber officinale* Roscoe	Ginger	Zingiberceae	Rhizomes

**Table 2 animals-13-00702-t002:** Degradability and gas production of *Themeda triandra* hay as influenced by the inclusion of plant extracts rich in phytochemicals and monensin at 16 h of fermentation in vitro.

Treatments	Apdeg (mg g^−1^ DM)	Trdeg (mg g^−1^ DM)	NDFD (mg g^−1^ DM)	MY (mg g^−1^ DM)	GP (mL g^−1^ DM)
*Acacia nilotica*	122 ^ab^	437 ^ab^	176 ^ab^	315	41 ^abcd^
*Acacia nilotica* (pods)	146 ^ab^	437 ^ab^	176 ^ab^	291	23 ^bcd^
*Acacia sieberiana*	191 ^a^	479 ^ab^	218 ^ab^	288	49 ^ab^
*Allium cepa*	214 ^a^	490 ^ab^	229 ^ab^	277	40 ^abcd^
*Allium sativum*	233 ^a^	460 ^ab^	199 ^ab^	227	42 ^abc^
*Aloe ferox*	201 ^a^	499 ^ab^	238 ^ab^	298	41 ^abcd^
*Ananas comosus*	201 ^a^	490 ^ab^	229 ^ab^	289	46 ^abc^
*Camellia japonica*	232 ^a^	488 ^ab^	227 ^ab^	256	46 ^abc^
*Carica papaya*	171 ^ab^	482 ^ab^	221 ^ab^	311	41 ^abcd^
*Carya illinoinensis*	−16 ^b^	407 ^b^	146 ^b^	423	6 ^d^
*Cichorium intybus*	239 ^a^	458 ^ab^	197 ^ab^	220	62 ^a^
*Citrus limon*	222 ^a^	492 ^ab^	231 ^ab^	269	51 ^ab^
*Coffea arabica*	119 ^ab^	475 ^ab^	214 ^ab^	356	46 ^abc^
*Ficus benjamina*	137 ^ab^	465 ^ab^	204 ^ab^	328	33 ^abcd^
*Ficus natalensis*	174 ^ab^	483 ^ab^	222 ^ab^	309	43 ^abc^
*Moringa oleifera*	151 ^ab^	493 ^ab^	232 ^ab^	341	46 ^abc^
*Morus nigra*	243 ^a^	498 ^ab^	237 ^ab^	254	42 ^abc^
*Persea Americana*	135 ^ab^	480 ^ab^	219 ^ab^	345	38 ^abcd^
*Psidium guajava*	148 ^ab^	447 ^ab^	186 ^ab^	299	33 ^abcd^
*Tulbaghia violacea*	161 ^ab^	511 ^a^	250 ^a^	350	36 ^abcd^
*Vernonia amygdalina*	204 ^a^	481 ^ab^	220 ^ab^	277	50 ^ab^
*Zingiber officinale*	212 ^a^	474 ^ab^	213 ^ab^	262	48 ^ab^
Monensin sodium	146 ^ab^	452 ^ab^	191 ^ab^	306	13 ^cd^
Control	255 ^a^	490 ^ab^	229 ^ab^	235	42 ^abc^
MSD	198.41	96.93	96.93	246.83	34.98
RMSE	62.89	30.72	30.72	78.24	11.09
Treatment effect	**	*	*	NS	***

Apdeg—apparent degradability, Trdeg—true degradability, NDFD—neutral detergent degradability, MY—microbial yield, GP—gas produced, MSD—a minimum significant difference, RMSE—root mean square error. Means within a column with the same superscript are not significantly different (*p* ≤ 0.05). NS—not significant (*p* > 0.05), * (*p* ≤ 0.05), ** (*p* < 0.01), *** (*p* < 0.0001).

**Table 3 animals-13-00702-t003:** Degradability and gas production of Themeda triandra hay as influenced by the inclusion of plant extracts rich in phytochemicals and monensin at 48 h of fermentation in vitro.

Treatments	Apdeg (mg g^−1^ DM)	Trdeg (mg g^−1^ DM)	NDFD (mg g^−1^ DM)	MY (mg g^−1^ DM)	GP (mL g^−1^ DM)
*Acacia nilotica*	313 ^a^	540 ^ab^	279 ^ab^	227 ^ab^	89 ^abcdefgh^
*Acacia nilotica* (pods)	278 ^a^	533 ^ab^	272 ^ab^	255 ^ab^	53 ^fgh^
*Acacia sieberiana*	383 ^a^	556 ^ab^	295 ^ab^	173 ^b^	98 ^abcdefg^
*Allium cepa*	376 ^a^	554 ^ab^	293 ^ab^	177 ^b^	117 ^abcde^
*Allium sativum*	319 ^a^	562 ^ab^	301 ^ab^	243 ^ab^	130 ^ab^
*Aloe ferox*	322 ^a^	518 ^ab^	257 ^ab^	196 ^b^	101 ^abcdef^
*Ananas comosus*	344 ^a^	552 ^ab^	291 ^ab^	208 ^b^	87 ^abcdefgh^
*Camellia japonica*	263 ^a^	552 ^ab^	291 ^ab^	290 ^ab^	64 ^efgh^
*Carica papaya*	289 ^a^	587 ^a^	326 ^a^	298 ^ab^	118 ^abcde^
*Carya illinoinensis*	46 ^b^	484 ^b^	223 ^b^	438 ^a^	62 ^defgh^
*Cichorium intybus*	304 ^a^	564 ^ab^	303 ^ab^	260 ^ab^	123 ^abc^
*Citrus limon*	385 ^a^	558 ^ab^	297 ^ab^	173 ^b^	140 ^a^
*Coffea arabica*	322 ^a^	533 ^ab^	272 ^ab^	212 ^b^	127 ^ab^
*Ficus benjamina*	304 ^a^	556 ^ab^	295 ^ab^	252 ^ab^	100 ^abcdefg^
*Ficus natalensis*	318 ^a^	561 ^ab^	300 ^ab^	242 ^ab^	104 ^abcdef^
*Moringa oleifera*	253 ^a^	545 ^ab^	284 ^ab^	292 ^ab^	109 ^abcdef^
*Morus nigra*	338 ^a^	528 ^ab^	267 ^ab^	190 ^b^	44 ^gh^
*Persea Americana*	325 ^a^	552 ^ab^	291 ^ab^	227 ^ab^	40 ^h^
*Psidium guajava*	238 ^ab^	499 ^ab^	238 ^ab^	261 ^ab^	92 ^abcdefgh^
*Tulbaghia violacea*	310 ^a^	530 ^ab^	269 ^ab^	220 ^ab^	119 ^abcde^
*Vernonia amygdalina*	205 ^ab^	549 ^ab^	288 ^ab^	344 ^ab^	121 ^abcd^
*Zingiber officinale*	296 ^a^	567 ^ab^	306 ^ab^	271 ^ab^	117 ^abcde^
Monensin sodium	327 ^a^	561 ^ab^	300 ^ab^	234 ^ab^	66 ^cdefgh^
Control	382 ^a^	532 ^ab^	271 ^ab^	151 ^b^	104 ^abcdef^
MSD	196.79	92.48	92.48	220.78	58.14
RMSE	62.38	29.31	29.31	69.98	18.43
Treatment effect	***	*	*	**	***

Apdeg—apparent degradability, Trdeg—true degradability, NDFD—neutral detergent degradability, MY—microbial yield, GP—gas produced, MSD—a minimum significant difference, RMSE—root mean square error. Means within a column with the same superscript are not significantly different (*p* ≤ 0.05). * (*p* ≤ 0.05), ** (*p* < 0.01), *** (*p* < 0.0001).

**Table 4 animals-13-00702-t004:** Effect of plant extracts and monensin on ruminal pH and partitioning factor at 16 h and 48 h of incubation in vitro.

	16 h Incubation		48 h Incubation	
Treatments	pH	PF	pH	PF
*Acacia nilotica*	6.89 ^ab^	11.7 ^c^	6.72 ^ab^	6.14 ^cdef^
*Acacia nilotica* (pods)	6.89 ^ab^	19.7 ^c^	6.74 ^ab^	10.52 ^abc^
*Acacia sieberiana*	6.87 ^ab^	10.1 ^c^	6.73 ^ab^	6.13 ^cdef^
*Allium cepa*	6.83 ^ab^	13.5 ^c^	6.64 ^ab^	4.83 ^ef^
*Allium sativum*	6.82 ^ab^	11.0 ^c^	6.67 ^ab^	4.46 ^ef^
*Aloe ferox*	6.84 ^ab^	12.2 ^c^	6.78 ^ab^	5.31 ^def^
*Ananas comosus*	6.85 ^ab^	10.9 ^c^	6.68 ^ab^	6.50 ^cdef^
*Camellia japonica*	6.85 ^ab^	10.8 ^c^	6.72 ^ab^	9.79 ^abcd^
*Carica papaya*	6.88 ^ab^	12.3 ^c^	6.72 ^ab^	5.08 ^ef^
*Carya illinoinensis*	6.96 ^a^	66.7 ^a^	6.91 ^a^	8.23 ^bcdef^
*Cichorium intybus*	6.84 ^ab^	7.8 ^c^	6.66 ^ab^	4.72 ^ef^
*Citrus limon*	6.85 ^ab^	10.0 ^c^	6.78 ^ab^	4.04 ^f^
*Coffea arabica*	6.90 ^ab^	10.7 ^c^	6.76 ^ab^	4.28 ^ef^
*Ficus benjamina*	6.92 ^ab^	15.6 ^c^	6.72 ^ab^	5.63 ^def^
*Ficus natalensis*	6.81 ^b^	11.7 ^c^	6.79 ^ab^	5.46 ^def^
*Moringa oleifera*	6.85 ^ab^	11.8 ^c^	6.58 ^b^	5.13 ^def^
*Morus nigra*	6.85 ^ab^	11.9 ^c^	6.75 ^ab^	12.56 ^ab^
*Persea Americana*	6.87 ^ab^	12.9 ^c^	6.83 ^ab^	14.32 ^a^
*Psidium guajava*	6.91 ^ab^	14.4 ^c^	6.70 ^ab^	5.47 ^def^
*Tulbaghia violacea*	6.80 ^b^	17.1 ^c^	6.62 ^b^	4.62 ^ef^
*Vernonia amygdalina*	6.85 ^ab^	10.5 ^c^	6.64 ^ab^	4.59 ^ef^
*Zingiber officinale*	6.84 ^ab^	11.4 ^c^	6.65 ^ab^	4.84 ^ef^
Monensin sodium	6.96 ^a^	36.7 ^b^	6.79 ^ab^	8.80 ^bcde^
Control	6.88 ^ab^	11.1 ^c^	6.78 ^ab^	5.12 ^def^
MSD	0.14	14.3	0.29	4.71
RMSE	0.05	4.5	0.09	1.49
Treatment effect	**	***	*	***

PF—partitioning factor, MSD—a minimum significant difference, RMSE—root mean square error. Means within a column with the same superscript are not significantly different (*p* ≤ 0.05). * (*p* ≤ 0.05), ** (*p* < 0.01), *** (*p* < 0.0001).

## Data Availability

Data is contained within the article.

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
