# Peer review of "Evaluating the Effects of Some Selected Medicinal Plant Extracts on Feed Degradability, Microbial Protein Yield, and Total Gas Production In Vitro"

_animals, 2023, doi:10.3390/ani13040702_

Round 1
Reviewer 1 Report
Manuscript ID: animals-2115696
Title: Evaluating the effects of some selected medicinal plant extracts on
feed degradability, microbial protein yield, and total gas production in vitro
Authors: Aderonke Naimot Abd'Quadri-Abojukoro *, Ignatius Verla Nsahlai
Submitted to section: Animal System and Management,
This study evaluates the effect of crude ethanolic plant
extracts on in vitro rumen fermentation of Themeda triandra hay using
monensin sodium as a positive control. Even though this work is interesting, several concerns occurred throughout the manuscript and required the authors to revise before further consideration.
Summary:
This section did not well reflect the main point of the work. Simply the result was not observed in this section.
Abstract :
It is so surfaces summarized and the significant result did not satisfactorily explain. What is experimental design?
L21: How does Themeda triandra hay relates to the title? Why did this hay reqired to focus?
L28-30: The authors mentioned that “plant extracts influenced…..” What exactly is effect? Increase? Decreased? Please specify.
L29: What does “ruminal degradation”? Is it mean “ruminal feed degradation”
L30: The authors investigated 22 plant species but cannot define what plant is the best! Thus, I suggest that suggesting the better one should be indicated.
L31: Present work was tested in in vitro, thus unsuitable to summarize over the scope of the topic and mention animal performance.
Introduction:
-This section is lack of literature review regarding 22 plant species in which the authors are interested. How do those plants come? L85-87, the authors provide some previous reports on plant extract used to modify rumen while the research gap of these works is not defined.
-The main criticism of this work is the lack of a hypothesis and the novelty of the work then not clear.
-In the abstract, the authors mention “Themeda triandra hay” but the introduction has no detail about hay!!!
L35-39, 65-70: Required citation.
L35-40: It seems to be out of the scope of recent work. The authors should narrow down better than the current form.
Materials and method
-How are the author's selected plant materials used in this study? Lack of detail of each plant's use eg age, collecting protocol, sample preparation, etc.
-Too rough of preparation of plant extracts, the authors should provide in brief for the specific method.
-Experimental design must be addressed.
Result
L197-198: Avoid repeating the report of the data in Table. Check for all parts.
Table 4:
-Why does ruminal pH at 16 h showed higher than 48 h incubation?? Normally, at earlier fermentation, pH should be lower due to high microbial activity to digest feed. Please discuss!
-Why at 16 h incubation, PF value was highest, whereas not observed in 48 h incubation? No discussion was observed in the section.
Discussion:
L330-336: Gas production in C. illinoinensis is only 6 ml/g DM which what lowest among plants! The author's defense is that high condensed tannins may be low GP. It was unbelievable when 6 mg was produced while another plant is also contained CT but not the lowest GP. There is any possible biological mechanism inside?
L408-409: How is monensin's ability to improve rumen fermentation and feed efficiency? Please provide a biological mechanism!
L411-412: Is it correct? Some plant produces a few GP!!
Conclusions
-Some of them summarized out-of-scope work. No animal performance is not evaluated. This section should simplify what kind of plants are suitable for further application.
Reference:
-Check the guidelines of the journal!!
-Citation in text should be indicated by numbering!!
Author Response
Dear sir/ma,
Please see attachment

Reviewer 2 Report
Reviewer’s comments on the manuscript by Abd’quadri-Abojukoro and Nsahlai. entitled: Evaluating the effects of some selected medicinal plant extracts on feed degradability, microbial protein yield, and total gas production in vitro.
Manuscript ID: animals-2115696
January, 2023.
Specific Comments that are listed for the different section below:
in vitro, italic in the entire manuscript.
Table 1: I am confuse the list of plant species, how about the chemical composition?
L140-147: One ML of the 22 crude plant extracts (5% w/w of substrate mass) was pipetted into each bottle separately. Please explain the why add 1 mL plant extract? And how about the concentration? Do you detect or not? Please clear.
L143-144: For Monensin, Do your country prohibit it. I suggest you change it if prohibit.
L155-156: In general, the incubation periods are 24 h and 48 h, why you select 16 and 48 h? Moreover, for roughage, 72 h incubation in in vitro rumen fermentation, and why you just stop incubation at 48 h?
L157-161: Total gas production was determine, where is gas volume data in this manuscript? Just 16 h and 48 h? you need to explain it. I suggest you add total gas production data in different time till 72 h.
Author Response
Dear sir/ma,
Please see attachment

Reviewer 3 Report
This work explored the effect of 22 crude ethanolic plant extracts on in vitro rumen fermentation of Themeda triandra hay. The authors found that plant extracts influenced gas production in a varied way relative to control at both hours of incubation. All plant extracts have higher microbial protein yield as compared to the control group at 48 h. The authors speculated that all the plant extracts improve the microbial protein yield, which is the major source of amino acids to ruminants and has significant importance to animal performance. The manuscript is in general well written and the method development process is described in detail. However, the following issues have to be addressed before this manuscript is suitable for publication.
1. In the section Discussion, line 355, more information about the bioactivity of saponins should be provided and discussed. The following references are suitable to be cited.
https://doi.org/10.1016/j.phymed.2022.154580
https://doi.org/10.1080/10408398.2018.1514580
https://doi.org/10.1016/j.fct.2021.112075
2. The format of the section References should be checked.
Ref. 1, "Journal of Applied Animal Research" should be italic.
Ref. 4, "Gas Production, Digestibility and Efficacy of Stored or Fresh Plant Extracts to Reduce Methane Production on Different Substrates" should be "Gas production, digestibility and efficacy of stored or fresh plant extracts to reduce methane production on different substrates".
Ref. 34, "BMC veterinary research" should be "BMC Veterinary Research".
Author Response
Dear sir/ma,
Please see attachment

Round 2
Reviewer 1 Report
The comments have been replied to clearly and suggested to accept in the present form. Congratulations!
Reviewer 2 Report
The authors have been revised according to my suggestion.